# Numerical Modeling on the Compliance and Load Capacity of a Two-Row Aerostatic Journal Bearing with Longitudinal Microgrooves in the Inter-Row Zone

Vladimir Kodnyanko *, Stanislav Shatokhin, Andrey Kurzakov, Yuri Pikalov, Iakov Pikalov, Olga Grigorieva, Lilia Strok and Maxim Brungardt

Polytechnic Institute, Siberian Federal University, 660079 Krasnoyarsk, Russia; SShatochin@sfu-kras.ru (S.S.); AKurzakov@sfu-kras.ru (A.K.); YAPikalov@sfu-kras.ru (Y.P.); YPikalov@sfu-kras.ru (I.P.); OGrigorieva@sfu-kras.ru (O.G.); LStrok@sfu-kras.ru (L.S.); MBrungardt@sfu-kras.ru (M.B.)
* Correspondence: VKodnyanko@sfu-kras.ru

**Abstract:** Aerostatic bearings are attractive, with minimal friction losses, high durability, and environmental friendliness. However, such designs have a number of disadvantages, including low load-bearing capacity and high compliance due to high air compressibility and limited injection pressure. The article proposes a double-row aerostatic journal bearing with an external combined throttling system and longitudinal microgrooves in the inter-row zone. It is hypothesized that the use of microgrooves will reduce the circumferential flows of compressed air, as a result of which the compliance should decrease and the bearing capacity should increase. To test the hypothesis, we carried out the mathematical modeling, calculations, and theoretical study of stationary operation modes of the bearing for small shaft eccentricities in the vicinity of the central equilibrium position of the shaft and bearing capacity for arbitrary eccentricities. Formulas were obtained for the numerical evaluation of compliance for bearings with a smooth bushing surface and with longitudinal microgrooves. Iterative finite-difference methods for evaluating the fields of the squared pressure are proposed, on the basis of which the load capacity of the bearings is calculated. Experimental verification of the bearing's theoretical characteristics was carried out, which showed satisfactory agreement between the compared data. The study of the compliance and load capacity of a microgroove bearing yielded impressive results. We show that the positive effect from the application of the improvement begins to manifest itself already at four microgrooves; the effect becomes significant at six microgrooves, and at twelve or more microgrooves, the circumferential flows in the bearing gap practically disappear; therefore, the bearing characteristics can be calculated on the basis of one-dimensional models of air lubrication longitudinal flow. Calculations have shown that for a length of $L = 1$, the maximum load capacity of a bearing with microgrooves is 1.5 times higher than that of a conventional bearing; for $L \geq 1.5$, the bearing capacity increases twice or more. The result obtained allows us to recommend the proposed improvement for practical use in order to increase the load capacity of aerostatic journal bearings significantly.

**Keywords:** aerostatic journal bearing; compliance; load capacity; longitudinal microgrooves

## 1. Introduction

Recently, there has been an increased interest in the use of aerostatic bearings in machines, particularly in machine tools and instruments. A large number of theoretical and experimental works have been published on various aspects of gas lubrication [1–8]. The greatest demand for aerostatic bearings is observed in machine-tool construction when using aerostatic bearings in the spindle units of metalworking machines [9–12].

Aerostatic bearings are attractive because of their main advantages, namely, low air viscosity, minimal friction losses, and, as a consequence, high durability and environmental friendliness [13–16].

At the same time, aerostatic bearings have a number of disadvantages, the most predominant of which are low load-bearing capacity, high compliance (low stiffness), and an increased tendency for instability. The first two drawbacks are explained by the limited injection pressure, and the last drawback is determined by the high compressibility of air, even in microvolumes [17,18]. To reduce compliance, aerostatic bearings are used with active compensation of airflow and with displacement compensators of the movable element (shaft) [19–24]. In such bearings, the problem of high compliance is solved because these types of bearings allow a decrease in compliance to zero and even negative values (in the latter case, the shaft relative to the housing is displaced in the direction opposite to the current load), which allows the structures to be used not only as supports but also as active deformation compensators of the technological system of machine tools in order to reduce the time spent and increase metalworking accuracy. A constructive innovation— the system of external combined throttling (SECT) [18]—provides a significant decrease in compliance and guarantees stability. The use of SECT can significantly increase the performance speed of bearings and improve their dynamic quality, significantly reducing the oscillation of transient processes caused by external forces, which also affects the accuracy of metalworking.

Progress towards improving compliance and stability characteristics has not, however, touched on one of the main performance characteristics of aerostatic bearings, namely, their bearing capacity. The low load capacity is not only impacted by the limited injection pressure, as there is another factor that only applies to radial aerostatic bearings. This factor relates to the presence of circumferential compressed air crossflows in the bearing gap of structures. Moreover, the longer the bearing, the greater these flows and, consequently, the lower the bearing capacity. In this paper, a method for increasing the bearing capacity by partially or almost completely suppressing the negative effect of circumferential gas lubricant crossflows in the bearing lubrication gap is considered.

However, we will first consider the characteristics of a conventional SECT aerostatic bearing since the improvements in the suppression of circumferential flows will be applied to these types of structures.

## 2. Characteristics of a Conventional Aerostatic Bearing

Figure 1 shows the design diagram of the structure.

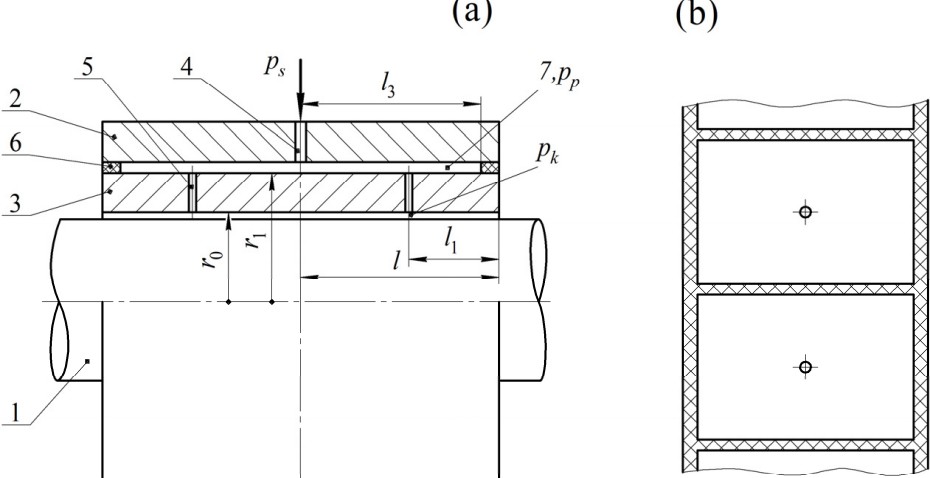

**Figure 1.** Design diagram of an aerostatic journal bearing with SECT: (**a**) longitudinal section, (**b**) unfolded drawing.

An aerostatic journal bearing (Figure 1a) consists of a shaft 1 (Figure 1), a body 2, and a stationary inner sleeve 3 resting on a rigid annular suspension 6. In the body 3, throttling diaphragms 4 are made, and, in the movable sleeve 3, there are damping annular

diaphragms 5, where resistance in a stable bearing should be at approximately an order of magnitude lower than the resistance of the feeders 4. This ensures maximum stability and optimal dynamic quality [18]. The surfaces of the body, bushing, and suspension form inter-throttle chambers 7, where volume plays an essential role in ensuring the optimal dynamic quality of the structure. In order to reduce the negative effect of the circumferential crossflow of air in the inter-throttle cavities on the load characteristics of the bearing, they are separated from each other by a sufficient number of longitudinal baffles, as shown in Figure 1b.

Compressed air under injection pressure $p_s$ = const (pumping source is not shown in Figure 1), overcoming the resistance of throttling feeders 4, enters the inter-throttling cavities 7 under pressure $p_p < p_s$. Further, through damping annular diaphragms 5, air enters the bearing gap under pressure $p_k < p_p$ and then flows out into the environment with pressure $p_a$.

Let us consider two mathematical models of the bearing's stationary state—a small movement of the shaft 1 in the vicinity of its central equilibrium position and an arbitrary position at any shaft eccentricity. We will investigate the characteristics under the assumption that the axes of the bushing and bearing are parallel.

## 3. General Mathematical Model of the Static State of a Conventional Bearing

We will consider, in a stationary mode, small radial displacements of the shaft 1 caused by small effects from the external load $f$.

Let us assume that the number of annular diaphragms in each row is large enough, which makes it possible to calculate the characteristics of the bearing using the method of continuous pressurization lines, which allows the replacement of discrete pressurization holes in each row with an equivalent continuous feeder with equal pressure at its outlet while maintaining the nature of the lubricant outflow from the diaphragms [25]. Studies show that such a model is trustworthy if the number of feeders $n_k > 15$ in each row [25]. In these cases, the error in calculating the performance of bearings with discrete feeders does not exceed 1%.

The study of the static characteristics of the bearing was carried out in a dimensionless form. The following are taken as scales of values: shaft radius $r_0$—for lengths, radii and longitudinal coordinate $z$; the thickness $h_0$ of the lubricating gap with the coaxial arrangement of the shaft and sleeve—for the current thickness $h$ of the bearing gap and eccentricity $e$; the ambient pressure $p_a$—for pressures; $2\pi r_0^2 p_a$—for forces; $\frac{\pi h_0^3 p_a^2}{6\mu R T n_d}$—for mass airflow rates, where $\mu$ is the coefficient of dynamic air viscosity, $R$ is the gas constant, $T$ is the absolute air temperature, and $n_d$ is the number of diaphragms in one row.

### 3.1. Boundary Value Problems for a Dimensionless Function of Pressure in the Bearing Gap

When the axes of the shaft 1 and sleeve 3 are parallel, the pressure function $P(Z,\varphi)$ in the bearing gap satisfies the stationary Reynolds equation [26]

$$\frac{\partial}{\partial \varphi}\left(H^3 P \frac{\partial P}{\partial \varphi}\right) + \frac{\partial}{\partial Z}\left(H^3 P \frac{\partial P}{\partial Z}\right) = 0, \tag{1}$$

where $Z$, $\varphi$ are longitudinal and circumferential coordinates.

$$H(\varphi) = 1 - \varepsilon \cos \varphi \tag{2}$$

is a function of the thickness of the bearing gap; $\varepsilon$ is an eccentricity.

To formulate boundary conditions for Equation (1), we introduced local coordinate systems. We will consider the symmetrical right half of the bearing. In the inter-row zone $0 \le Z \le L_2$, in the end zone $0 \le Z \le L_1$, and for both of these zones $0 \le \varphi \le 2\pi$, where $L_2 = L - L_1$ is half the length of the inter-row zone, $L_1$ is the length of the end zones, and $L$ is half the bearing length.

Let us take $\Psi = P^2$. Then, the corresponding boundary value problem for Equation (1) takes the form

$$
\begin{cases}
\frac{\partial}{\partial \varphi}\left(H^3 \frac{\partial \Psi}{\partial \varphi}\right) + \frac{\partial}{\partial Z}\left(H^3 \frac{\partial \Psi}{\partial Z}\right) = 0, \\
\frac{\partial \Psi}{\partial Z}(0, \varphi) = 0, \Psi(L_1, \varphi) = \Psi_k(\varphi), \\
\Psi(0, \varphi) = \Psi_k(\varphi), \Psi(L_2, \varphi) = 1, \\
\Psi(Z, \varphi) = \Psi(Z, \varphi + 2\pi), \\
\frac{\partial \Psi}{\partial \varphi}(Z, \varphi) = \frac{\partial \Psi}{\partial \varphi}(Z, \varphi + 2\pi),
\end{cases}
\tag{3}
$$

where $\Psi_k(\varphi) = P_k^2(\varphi)$ is the squared function of the pressure on the boost line at the air outlet from the damping annular diaphragms.

The response to small loads $F$ will be small changes in the bearing gap, eccentricity $\Delta\varepsilon$, and the square of the pressures at the feeders' outlet

$$
\Psi_k(\varphi) = \Psi_{k0} + \Delta\Psi_k \cos\varphi,
\tag{4}
$$

$$
\Psi_p(\varphi) = \Psi_{p0} + \Delta\Psi_p \cos\varphi,
\tag{5}
$$

where $\Psi_{k0} = P_{k0}^2$, $\Psi_{p0} = P_{p0}^2$ are the values of the function $\Psi$ at the exit from the annular and simple diaphragms with the coaxial arrangement of the shaft and sleeve, $\Delta\Psi_k, \Delta\Psi_p$ are small deviations.

By analogy, let us take

$$
\Psi(Z, \varphi) = \Psi_0(Z) + \Delta\Psi(Z) \cos\varphi.
\tag{6}
$$

Substituting (6) into (1) and performing the separation of functions, taking into account (4), we obtain the boundary value problem for the function $\Psi_0(Z)$

$$
\begin{cases}
\frac{d^2\Psi_0}{dZ^2} = 0, \\
\frac{d\Psi_0}{dZ}(0) = 0, \Psi_0(L_2) = \Psi_{k0}, \\
\Psi_0(0) = \Psi_{k0}, \Psi_0(L_1) = 1
\end{cases}
\tag{7}
$$

and the boundary value problem for the function of small deviations $\Delta\Psi(Z)$

$$
\begin{cases}
\frac{d^2\Delta\Psi}{dZ^2} - \Delta\Psi = 0, \\
\Delta\Psi(0) = \Delta\Psi_k, \Delta\Psi(L_1) = 0, \quad 0 \le Z \le L_1, \\
\frac{d\Delta\Psi}{dZ}(0) = 0, \Delta\Psi(L_2) = \Delta\Psi_k, \quad 0 \le Z \le L_2.
\end{cases}
\tag{8}
$$

The solution to Problem (7) is the function

$$
\Psi_0(Z) = \begin{cases}
\left(P_{k0}^2 - 1\right)\frac{L_1 - Z}{L_1} + 1, & 0 \le Z \le L_1, \\
P_{k0}^2, & 0 \le Z \le L_2.
\end{cases}
\tag{9}
$$

The solution to Problem (8) is the function

$$
\Delta\Psi(Z) = \begin{cases}
\Delta\Psi_k \frac{\operatorname{sh}(L_1 - Z)}{\operatorname{sh}L_1}, & Z \le L_1, \\
\Delta\Psi_k \frac{\operatorname{ch}Z}{\operatorname{ch}L_2}, & Z \le L_2.
\end{cases}
\tag{10}
$$

The dimensional bearing capacity of the bearing is calculated by the following formula [26]:

$$
w = 2\int_0^{2\pi}\int_0^L (p - p_a)\cos\varphi\, dz\, d\varphi.
\tag{11}
$$

After reducing (11) to a dimensionless form, we obtain

$$
\begin{aligned}
W &= \frac{2r_0^2 p_a}{2\pi r_0^2 p_a} \int_0^{2\pi} \int_0^L \left( \sqrt{\Psi_0 + \Delta\Psi \cos\varphi} - 1 \right) \cos\varphi \, dZ d\varphi = \\
&= \frac{1}{\pi} \int_0^{2\pi} \cos^2\varphi \, d\varphi \int_0^L \frac{\Delta\Psi}{2\sqrt{\Psi_0}} dZ = \frac{1}{2} \int_0^L \frac{\Delta\Psi}{\sqrt{\Psi_0}} dZ.
\end{aligned}
\tag{12}
$$

In Formula (12), we express $\Psi$ in terms of $P$. Since

$$
\Psi_0 + \Delta\Psi \cos\varphi = (P_0 + \Delta P \cos\varphi)^2 = P_0^2 + 2P_0 \cos\varphi,
$$

then

$$
\Psi_0 = P_0^2, \Delta\Psi = 2P_0\Delta P.
\tag{13}
$$

Substituting (13) into (12), we obtain

$$
\Delta W = \int_0^L \Delta P \, dZ.
\tag{14}
$$

Replacing $\Psi$ in (1) by $P$, we write (10) in the form

$$
\Delta P(Z) = \begin{cases} \Delta P_k \frac{P_{k0} \operatorname{sh}(L_1 - Z)}{P_0(Z) \operatorname{sh} L_1}, & Z \leq L_1, \\ \Delta P_k \frac{\operatorname{ch} Z}{\operatorname{ch} L_2}, & Z \leq L_2, \end{cases}
\tag{15}
$$

where, taking into account (9)

$$
P_0(Z) = \sqrt{\Psi_0(Z)} = \begin{cases} \sqrt{\left(P_{k0}^2 - 1\right) \frac{L_1 - Z}{L_1} + 1}, & 0 \leq Z \leq L_1, \\ P_{k0}, & 0 \leq Z \leq L_2. \end{cases}
\tag{16}
$$

Substituting (15) and (16) in (14), we obtain the formula for determining the small deviation of the bearing capacity

$$
\Delta W = A_w \Delta P_k,
\tag{17}
$$

where

$$
A_w = \operatorname{th} L_2 + \frac{P_{k0}}{\operatorname{sh} L_1} \int_0^{L_1} \frac{\operatorname{sh}(L_1 - Z)}{P_0(Z)} dZ.
$$

At the outlet of the pressurization line and the inlet into the bearing gap, the dimensionless gas flow rate $Q_h$ is distributed in two directions: $Q_{h2}$ into the inter-row zone and $Q_{h1}$ into the end zone [26]

$$
Q_h = Q_{h2} + Q_{h1},
\tag{18}
$$

where

$$
Q_{h1} = -H^3 \left( \frac{\partial\Psi}{\partial Z} \right)_{Z = L_1 - 0},
\tag{19}
$$

$$
Q_{h2} = H^3 \left( \frac{\partial\Psi}{\partial Z} \right)_{Z = L_1 + 0}.
\tag{20}
$$

Taking into account (4) and (13) and performing linearization (19), we obtain

$$
Q_{h10} = -\left( \frac{dP_0^2}{dZ} \right)_{Z = 0},
\tag{21}
$$

and

$$\Delta Q_{h1} = \left[ -2\frac{d(P_0\Delta P)}{dZ} + 3\frac{dP_0^2}{dZ}\Delta\varepsilon \right]_{Z=0}. \tag{22}$$

Having performed the differentiation in (15) and (16), we describe the formula for the flow rate in the end zone with the central position of the shaft as

$$Q_{h10} = \frac{P_{k0}^2 - 1}{L_1} \tag{23}$$

and the formula for the small deviation of the flow rate in this part as

$$\Delta Q_{h1} = A_{qp1}\Delta P_k - A_{qe1}\Delta\varepsilon, \tag{24}$$

where

$$A_{qp1} = 2P_{k0}\text{th}L_1, A_{qe1} = 3Q_{h10}.$$

Similarly, for the inter-row zone, we obtain

$$\begin{cases} Q_{h20} = 0, \\ \Delta Q_{h2} = A_{qp2}\Delta P_k, \end{cases} \tag{25}$$

where

$$A_{qp2} = 2P_{k0}\text{cth}L_2. \tag{26}$$

Substituting (23)–(26) into (18), we find the final formulas for the separated components of the air flow rates in the bearing gap

$$Q_{h0} = \frac{P_{k0}^2 - 1}{L_1}, \tag{27}$$

$$\Delta Q_h = A_{qp}\Delta P_k - A_{qe}\Delta\varepsilon, \tag{28}$$

where

$$A_{qp} = 2P_{k0}(\text{cth}L_1 + \text{th}L_2), \tag{29}$$

$$A_{qe} = 3Q_{h0}. \tag{30}$$

To determine the mass flow rate of air through the diaphragms, we used the formula obtained by integrating the boundary value problem for the nonlinear Bernoulli Equation [26]

$$q = \sqrt{\frac{2}{RT}\frac{\gamma}{\gamma-1}}s \cdot \text{Brn}(p_1, p_2), \tag{31}$$

where

$$\text{Brn}(p_1, p_2) = p_1 \begin{cases} C_\gamma, & \frac{p_2}{p_1} \leq C_c, \\ \sqrt{\left(\frac{p_2}{p_1}\right)^{\frac{2}{\gamma}} - \left(\frac{p_2}{p_1}\right)^{\frac{\gamma+1}{\gamma}}}, & \frac{p_2}{p_1} > C_c, \end{cases} \tag{32}$$

and where $p_1, p_2$ are the pressures at the inlet and outlet of the diaphragms; $p_1 \geq p_2$, $s$ is the effective part of the air outflow from the diaphragms ($s = \pi dh$ for annular diaphragms, $s = \frac{\pi d^2}{4}$ for simple diaphragms); and $C_c = \frac{p_2}{p_1} = \left(\frac{2}{\gamma+1}\right)^{\frac{\gamma}{\gamma+1}} \approx 0.528$ is the critical pressure ratio at the adiabatic exponent for air $\gamma = 1.4$ [26].

After reduction to a dimensionless form, we obtain the formula for the flow rate through the annular diaphragms

$$Q_k = A_k H \text{Brn}(P_p, P_k), \tag{33}$$

and the formula for the flow-through simple diaphragms

$$Q_p = A_p \operatorname{Brn}(P_s, P_p),$$ (34)

where

$$A_k = \frac{6\mu\gamma_1 n_d d\sqrt{RT}}{h_0^2 p_a}, \ A_p = \frac{3\mu\gamma_1 d^2 n_d \sqrt{RT}}{2h_0^3 p_a}, \ \gamma_1 = \sqrt{\frac{2\gamma}{\gamma-1}} \approx 2.646.$$

*3.2. System of Equations for the Balance of Forces and Air Flow Rate in the SECT*

The dimensionless system of equations describing the static equilibrium of the bearing includes one equation for forces and two equations for the balance of air flow rates in the SECT

$$\begin{cases} W = F, \\ Q_p - Q_k = 0, \\ Q_k - Q_h = 0. \end{cases}$$ (35)

It is convenient to calculate the constant static pressure $P_{k0}$ at the outlet of the damping feeders and the corresponding pressure $P_{p0}$ in the inter-throttling chambers using the normalized coefficients [18,21,24]

$$\chi = \frac{P_{k0}^2 - 1}{P_s^2 - 1} \in [0,1], \quad \varsigma = \frac{P_{p0}^2 - P_{k0}^2}{P_s^2 - P_{k0}^2} \in [0,1].$$ (36)

By setting the injection pressure $P_s$, $\chi$ and $\varsigma$, we can find the dimensionless pressures of the unloaded bearing

$$P_{k0} = \sqrt{1 + \chi(P_s^2 - 1)}, \quad P_{p0} = \sqrt{P_{k0}^2 + \varsigma(P_s^2 - P_{k0}^2)}.$$ (37)

With the help of (27), it is now possible to calculate the flow rate in the bearing gap. Using (35)–(37) at $\varepsilon = 0$, we find the criteria for the similarity of damping annular diaphragms and throttling simple diaphragms

$$A_k = \frac{Q_{h0}}{\Pi(P_{p0}, P_{k0})}, \quad A_p = \frac{Q_{h0}}{\Pi(P_s, P_{p0})}.$$ (38)

After linearizing (33) and (34), we find the deviations of the flow rates through the diaphragms

$$\Delta Q_k = A_{qkk}\Delta P_k + A_{qkp}\Delta P_p - A_{qke}\Delta\varepsilon,$$ (39)

$$\Delta Q_p = A_{qpp}\Delta P_p,$$ (40)

where

$$A_{qkk} = \frac{\partial \operatorname{Brn}(P_{p0}, P_{k0})}{\partial P_{k0}}, \quad A_{qkp} = \frac{\partial \operatorname{Brn}(P_{p0}, P_{k0})}{\partial P_{p0}},$$
$$A_{qke} = \operatorname{Brn}(P_{p0}, P_{k0}), \ A_{qpp} = \frac{\partial \operatorname{Brn}(P_s, P_{p0})}{\partial P_{p0}}.$$

From (35), it follows that the equations of forces and flow rates for deviations can be written in the form

$$\begin{cases} \Delta W = \Delta F, \\ \Delta Q_p - \Delta Q_k = 0, \\ \Delta Q_k - \Delta Q_h = 0. \end{cases}$$ (41)

Substituting (17), (27), (39), and (40) into (41), we obtain a system of linear equations, which can be written in matrix form as

$$\begin{bmatrix} 0 & A_w & 0 \\ A_{qke} & -A_{qkk} & (A_{qpp} - A_{qkp}) \\ (A_{qe} - A_{qke}) & (A_{qkk} - A_{qp}) & A_{qkp} \end{bmatrix} \begin{bmatrix} K_e \\ K_k \\ K_p \end{bmatrix} = \begin{bmatrix} 1 \\ 0 \\ 0 \end{bmatrix},$$ (42)

where $K_e = \frac{\Delta \varepsilon}{\Delta F}$, $K_k = \frac{\Delta P_k}{\Delta F}$, $K_p = \frac{\Delta P_p}{\Delta F}$ are the gains of the transfer functions, which are equal to the ratio of the output deviations $\Delta \varepsilon$, $\Delta P_k$, $\Delta P_p$ and the small input force deviation $\Delta F$. The quantity $K_e$ is the static compliance of the bearing.

Having solved the system of (42), we find the static compliance of the bearing as

$$K_e = \frac{1}{A_w} \frac{\left(A_{qp} - A_{qkk}\right)\left(A_{qpp} - A_{qkp}\right) - A_{qkk}A_{qkp}}{\left(A_{qe} - A_{qke}\right)\left(A_{qpp} - A_{qkp}\right) - A_{qke}A_{qkp}}. \tag{43}$$

The resulting formula makes it possible to evaluate the effect of circumferential flows on bearing compliance at small shaft displacements.

Figure 2 shows the dependences of the compliance $K_e$ on the length $L$ at various values of the relative elongation of the end length $\lambda_1 = L_1/L$, the injection pressure $P_s = 5$, and the recommended values of the SECT tuning factors $\chi = 0.45$, $\varsigma = 0.15$ [18,21].

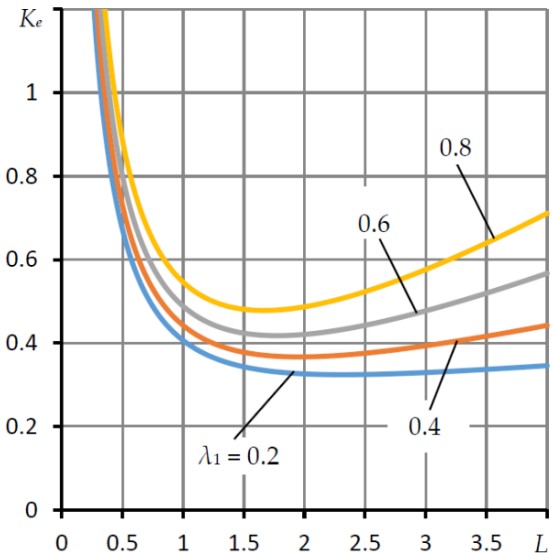

**Figure 2.** Dependences of the compliance $K_e$ of a lightly loaded bearing on the length $L$ at different values of the relative elongation of the end zone $\lambda_1$.

With an increase in $L$, the part of the bearing surface lubricated by air increases should contribute to a decrease in the compliance $K_e$. However, as the graphs show, a decrease in compliance occurs only up to a certain $L_{opt}$, which corresponds to the minimum compliance. For example, for $L = 1.5$ and $\lambda_1 = 0.4$, the minimum compliance of $K_e = 0.367$ occurs at $L = 1.97$. With a further increase in $L$, the compliance begins to increase. The reason for the deterioration of compliance, in this case, is significant air crossflows. Obviously, at small eccentricities, the bearing capacity will also decrease since $W = \frac{\varepsilon}{K_e}$. In this regard, the dependences of compliance and bearing capacity for arbitrary working eccentricities ($\varepsilon \leq 0.8$) are of interest.

### 3.3. Bearing Model with Arbitrary Eccentricities

We transform the Reynolds equation of Problem (3) and write it in the form

$$U\frac{\partial \Psi}{\partial \varphi} + \frac{\partial^2 \Psi}{\partial \varphi^2} + \frac{\partial^2 \Psi}{\partial Z^2} = 0, \tag{44}$$

where $\Psi(Z, \varphi) = P^2(Z, \varphi)$, $U(\varphi) = \frac{3H'}{H} = \frac{3\sin \varphi}{1 - \varepsilon \cos \varphi}$.

We divided the segment $\varphi \in [0, \pi]$ into an even number of $m$ parts and segment $Z \in [0, L]$ into an even number of $n$ parts. Let us write Equation (44) in a finite-difference form [27,28]

$$U_j \frac{\Psi_i^{j+1} - \Psi_i^{j-1}}{2\nu_\varphi} + \frac{\Psi_i^{j+1} - 2\Psi_i^j + \Psi_i^{j-1}}{\nu_\varphi^2} + \frac{\Psi_{i+1}^j - 2\Psi_i^j + \Psi_{i-1}^j}{\nu_z^2} = 0, \tag{45}$$

where $\nu_\varphi = \frac{\pi}{m}, \nu_z = \frac{L}{n}$ are the steps of integration with respect to the variables $\varphi$ and $Z$; $j = 1, 2, \ldots, m-1$, and $i = 1, 2, \ldots, k-1, k+1, \ldots, n-1$ are the numbers of the nodal points for these variables. For the injection line $i = k$, we composed separate equations.

We write the first boundary condition (3) in the following form [27,28]:

$$\frac{-\Psi_2^j + 4\Psi_1^j - 3\Psi_0^j}{2\nu_z} = 0. \tag{46}$$

The second boundary condition (3) for $j = 0, 2, ..., m$ is written in the following form:

$$\Psi_k^j = \left(P_k^2\right)^j, \tag{47}$$

where $k = \left[\frac{nL_2}{L}\right]$ is the number of the point on the grid of the longitudinal axis, which corresponds to the location of the pressurization line $Z = L_2$.

The fourth condition (3) for $j = 0, 1, ..., m$ is

$$\Psi_n^j = 1. \tag{48}$$

We write the last boundary conditions (3) for $i = 0, 1, ..., n$ in the form

$$\Psi_i^1 = \Psi_i^{-1}, \Psi_i^{m+1} = \Psi_i^{m-1}. \tag{49}$$

We represent (45) in the form

$$\left(b_0 + a_0 U_j\right)\Psi_i^{j+1} + c_0 \Psi_i^j + \left(b_0 - a_0 U_j\right)\Psi_i^{j-1} + \Psi_{i+1}^j + \Psi_{i-1}^j = 0, \tag{50}$$

where $a_0 = \frac{\nu_z^2}{2\nu_\varphi}, b_0 = \left(\frac{\nu_z}{\nu_\varphi}\right)^2, c_0 = -(1 + b_0)$.

Applying the well-known formulas of the quadratic order of accuracy to represent the first derivative of the function at the edges of the segment [27], we write the formulas for the flow rate in the bearing gap on the pressurization line as

$$Q_{h,j} = \frac{H_j^3}{2\nu_z}\left(\Psi_{k+2}^j - 4\Psi_{k+1}^j + 6\Psi_k^j - 4\Psi_{k-1}^j + \Psi_{k-2}^j\right). \tag{51}$$

Corresponding to (33) and (34), the finite-difference equations of the flow rate through the orifices have the form

$$Q_{k,j} = A_k H_j \operatorname{Brn}\left(P_p^j, P_k^j\right), \tag{52}$$

$$Q_{p,j} = A_p \operatorname{Brn}\left(P_s, P_p^j\right). \tag{53}$$

The final two equations in (35), as well as (52) and (53), give the system of equations

$$\frac{H_j^2}{2\nu_z}\left(\Psi_{k+2}^j - 4\Psi_{k+1}^j + 6\Psi_k^j - 4\Psi_{k-1}^j + \Psi_{k-2}^j\right) = A_k \operatorname{Brn}\left(P_p^j, P_k^j\right), \tag{54}$$

$$A_k H_j \operatorname{Brn}\left(P_p^j, P_k^j\right) = A_p \operatorname{Brn}\left(P_s, P_p^j\right). \tag{55}$$

The system of Equations (46)–(50), (54), and (55) was solved by the iteration method. For this, Equation (50) at intermediate nodes is represented in the form

$$\left\{\Psi_i^j\right\}_{\alpha+1} = -\frac{1}{c_0}\left\{(b_0 + a_0 U_j)\Psi_i^{j+1} + (b_0 - a_0 U_j)\Psi_i^{j-1} + \Psi_{i+1}^j + \Psi_{i-1}^j\right\}_{\alpha}, \tag{56}$$

where $j = 1, 2, \ldots, m-1$; $i = 1, 2, \ldots, k-1, k+1, \ldots, n-1$; $\alpha$ is the iteration number.

Similar iterative formulas on the pressurization line have the form

$$\left\{\Psi_k^j\right\}^{(\alpha+1)} = \frac{1}{6}\left\{\frac{2\nu_z}{H_j^2}A_k \operatorname{Brn}\left(P_p^j, P_k^j\right) - \left(\Psi_{k+2}^j - 4\Psi_{k+1}^j - 4\Psi_{k-1}^j + \Psi_{k-2}^j\right)\right\}^{(\alpha)}, \tag{57}$$

$$j = 0, 1, \ldots, m.$$

The pressures $P_p^j$ $(j = 0, 1, \ldots, m)$ at the new iteration were found by solving the nonlinear equations (55). For this, they were presented in the form

$$A_k H_j \operatorname{Brn}\left(\left\{P_p^j\right\}^{(\alpha+1)}, \left\{P_k^j\right\}^{(\alpha)}\right) - A_p \operatorname{Brn}\left(P_s, \left\{P_p^j\right\}^{(\alpha+1)}\right) = 0. \tag{58}$$

For $i = 0$, Formula (46) is used. For this, the iterative scheme has the form

$$\left\{\Psi_0^j\right\}^{(\alpha+1)} = \frac{1}{3}\left\{\Psi_2^j - 4\Psi_1^j\right\}^{(\alpha)}, \tag{59}$$

$$j = 0, 1, \ldots, m.$$

The values of Function (16) were taken as the initial value of the pressure fields at the nodal points.

Bearing load capacity

$$W = \frac{2}{\pi}\int_0^\pi\int_0^L (P-1)\cos\varphi \, dZ d\varphi \tag{60}$$

was determined by the Simpson cubature formula [29]. Calculations were performed for $n = 20$ and $m = 20$.

Figure 3 shows the graphs of the dependence of the load capacity $W$ on the eccentricity $\varepsilon$ at various values of the length $L$ at $\lambda_1 = 0.5$, $P_s = 5$, $\chi = 0.45$, $\varsigma = 0.15$.

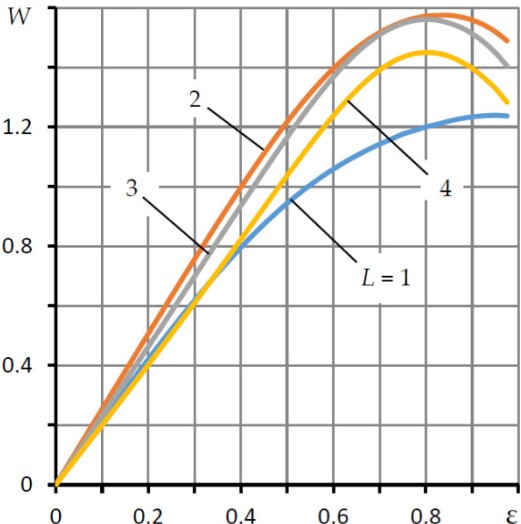

**Figure 3.** Dependences of the bearing load capacity $W$ on the eccentricity $\varepsilon$ for different values of the length $L$.

It can be seen from the graphs that in the region of small and moderate eccentricities, the dependence $W(\varepsilon)$ is close to linear and can be approximately represented by a simple formula

$$W(\varepsilon) \approx \frac{\varepsilon}{K_e},\qquad(61)$$

therefore, the bearing capacity is characterized by the same characteristic of bearing stiffness, which is the opposite of the compliance $K_e$. Based on this, it can be assumed that a decrease in circumferential air flows in the bearing gap will contribute not only to a decrease in compliance but also to an increase in bearing capacity, which, as mentioned above, is relevant for radial aerostatic bearings.

Further, a method for reducing circumferential flows and, therefore, reducing their effect on the compliance and bearing capacity of the bearing is considered.

## 4. Bearing with Longitudinal Microgrooves

The proposed improvement consists of the use of $n_g$ longitudinal microgrooves 3, which, as shown in Figure 4, are made on the bearing surface 1, at the same distance from each other.

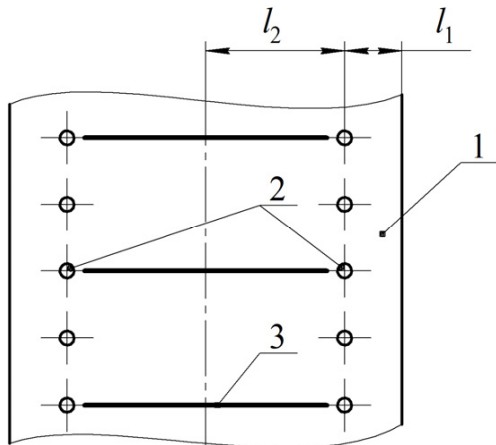

**Figure 4.** Unwrapping of the supporting surface with microgrooves.

The microgrooves are located in the inter-row zone at a minimum distance from the annular diaphragms 2, in order. Firstly, this is so as not to disrupt the nature of the outflow from the diaphragms, and, secondly, it is to provide approximately equal air pressures at the outlet from the diaphragm and the entrance to the adjacent microgroove. The idea is that due to the sufficient depth and width of the microgrooves and their number, it is practically possible to eliminate circumferential air flows and, thereby, reduce the compliance of the bearing gap and increase the bearing capacity.

### 4.1. Modeling the Effect of the Depth and Width of the Microgroove on Circumferential Crossflows

Let us perform a one-dimensional simulation of the compressed air flow between two parallel plates with a unit gap, one of which has a depression of width $L_D$ and depth $D$. The injection (supply) pressure $P_s$ is applied to the gap inlet; the pressure at the inlet to the depression is equal to $P_1$; at the outlet the pressure is $P_2$, and at the exit from the gap, the pressure is $P_a = 1$ (Figure 5).

The purpose of the study is to determine at what depth and width of the depression at its edges, and therefore throughout its entire length, the pressure practically does not change, which is evidence that there is nearly no resistance to airflow in the depression. This result will make it possible to determine at what depth and width of the microgroove of the bearing under study it can be assumed that the pressure upon it is constant and equal to the pressure at the outlet of the nearest diaphragm [30].

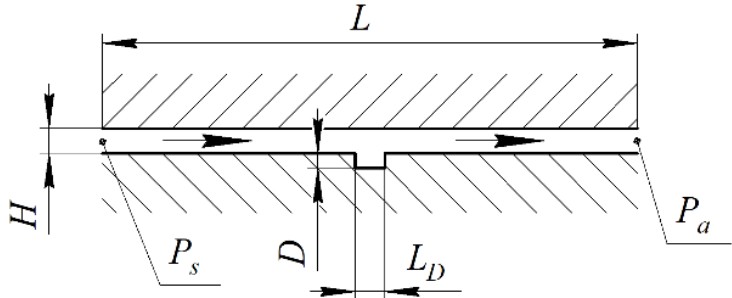

**Figure 5.** Scheme of compressed air flow in a constant gap with a depression.

Based on (9), with a constant clearance ($D = 0$), the pressure can be found by the formula

$$P(Z) = \sqrt{(P_s^2 - P_a^2)\frac{L - Z}{L} + P_a^2}.$$  (62)

If the depression is located in the middle of the gap and $D > 0$, then the local coordinate systems are

$$P(Z) = \begin{cases} \sqrt{(P_s^2 - P_1^2)\frac{L_H - Z}{L_H} + P_1^2}, & 0 \le Z \le L_H, \\ \sqrt{(P_1^2 - P_2^2)\frac{L_D - Z}{L_D} + P_2^2}, & 0 \le Z \le L_D, \\ \sqrt{(P_2^2 - P_a^2)\frac{L_H - Z}{L_H} + P_a^2}, & 0 \le Z \le L_H, \end{cases}$$  (63)

where $L_H = \frac{1}{2}(L - L_D)$.

Based on (62) and (63), we find the pressures $P_1$ and $P_2$ from the balance equations for the flow rates at the edges of the depression

$$\begin{cases} \frac{P_s^2 - P_1^2}{L_H} = (1 + D)^3 \frac{P_1^2 - P_2^2}{L_D}, \\ \frac{P_2^2 - P_a^2}{L_H} = (1 + D)^3 \frac{P_1^2 - P_2^2}{L_D}. \end{cases}$$  (64)

Having solved the system of the equation in (63), we obtain

$$\begin{cases} P_1 = \sqrt{\frac{P_s^2(B+1) + BP_a^2}{(B+1)^2 - B^2}}, \\ P_2 = \sqrt{\frac{P_a^2(B+1) + BP_s^2}{(B+1)^2 - B^2}}, \end{cases}$$  (65)

where $B = \frac{L_H}{L_D}(1 + D)^3$.

The ratio of the pressure difference at the edges of the step, calculated by Formula (62) at $D = 0$ and at $D \ge 0$ and calculated by formulas (64) at $D = L_D$, is

$$\Delta = \frac{P(L_H) - P(L_H + L_D)}{P_1 - P_2} \approx (1 + D)^3.$$  (66)

From Formula (65), it follows that the drop in resistance to airflow through the depression is proportional to the cube of the sum of the gap value and the size of the depression. At $D = 1.15$, the resistance drops by an order of magnitude, and, at $D = 3.64$, it drops by two orders of magnitude; therefore, we can assume that at $D \ge 4$, the resistance to the airflow in the depression and, consequently, in the microgrooves practically disappears. This allows us to conclude that if the depth of the microgroove is four times or more than the thickness of the lubricating gap, then the pressure in the microgroove can be approximately considered constant and equal to the air pressure at the outlet of the nearest annular diaphragm.

At the same time, it should be borne in mind that the presence of air volumes in the microgrooves can lead to a deterioration in the dynamic characteristics of the bearing. It is known that in order to maintain stability, it is necessary that additional volumes of air in contact with the bearing gap do not exceed its volume [17].

Let us make the necessary calculations. The volumes of the bearing gap and the microgrooves, respectively, are equal $v_h = 4\pi r_0 l h_0$ and $v_g = 2n_g(l - l_1)\delta^2$, where $\delta$ is the depth and width of the microgrooves. If we assume that the gap is three orders of magnitude smaller than the radius of the bearing surface $\frac{h_0}{r_0} = 10^{-3}$, then the ratio of the volumes of the microgrooves and the bearing gap will be $k_v = \frac{n_g D^2}{2000\pi}$ in excess. This ratio, as mentioned above, should not exceed $k_v = 1$. Taking $D = 4$ with a margin, we obtain the maximum allowable number of microgrooves $n_g = \frac{2000\pi}{16} \approx 400$. Obviously, the volume of microgrooves cannot exceed the volume of the bearing gap since the number of micro-grooves can be counted in units or tens (a more accurate estimate will be given below), but not in hundreds. Thus, the presence of microgrooves will not lead to a loss of bearing stability due to too much additional air volume in contact with the bearing gap.

### 4.2. Modeling Circumferential Crossflows in the Inter-Row Zone

Let us simulate circular air flows in the inter-row zone and estimate their effect on the load capacity of the air gap in this part, depending on the number of microgrooves, which are located in a circle at the same distance from each other on the lines $\varphi_k = \frac{2\pi(k-1)}{n_g}, k = 1, 2, \ldots, n_g$. We will assume that the number of microgrooves $n_g$ is an even number. Due to the symmetry of flows in both directions, we will consider the area as $G = \{0 \leq Z \leq L, 0 \leq \varphi \leq \pi\}$. We divide the area $G$ into sections $G_k = \{0 \leq Z \leq L, \varphi_k \leq \varphi \leq \varphi_{k+1}\}, k = 1, 2, \ldots, n_h$, where $n_h = 0.5n_g$ is the number of sections. Consider the boundary conditions for the square of the pressure in each section on the pressurization line and microgrooves

$$\Pi(\varphi) = P_{k0}^2(1 + \varepsilon \cos \varphi)^2. \tag{67}$$

For the boundary $Z = 0$, due to the symmetry of the flow, we will use the second condition of the boundary value problem (3).

We divide each section into $n \times m$ parts and perform a finite-difference replacement of the differential equation of Problem (3). In this case, we obtain an algebraic boundary value problem for the system of Equation (50) with boundary conditions (46), (49), and (66). We solve the problem using the iteration method, using Formula (56) for the interior points. For the boundary points of the region, we use Formulas (66) and (59). The load capacity of the compressed air gap in the $G$ region was found by summing the force reactions on the $G_k$ segments

$$W = \sum_{k=1}^{n_h} W_k.$$

In the absence of circumferential flows, $\Psi(Z, \varphi) = \Pi(\varphi)$, therefore

$$W = \frac{2P_{k0}}{\pi} \int_0^\pi \int_0^L (1 + \varepsilon \cos \varphi) \cos \varphi \, dZ d\varphi = P_{k0} L \varepsilon.$$

Figure 6 shows the dependences of the bearing capacity $W$ of the compressed air gap in the region $G$ on its length $L$ for a different number of microgrooves $n_g$ at $P_{k0} = 3$, $\varepsilon = 0.75$.

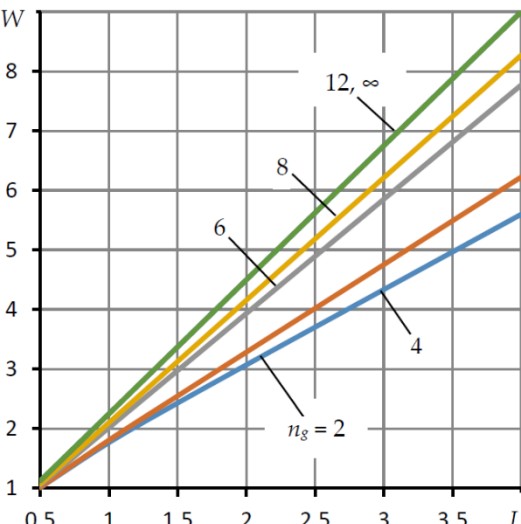

**Figure 6.** Dependences of the bearing capacity $W$ of the sectors on their length $L$ for various numbers of microgrooves $n_g$.

The graph shows that an increase in the number of grooves begins to noticeably affect the bearing capacity immediately. Already at $n_g$ = 4, in comparison with the graph for $n_g$ = 2, at which there are no barriers to crossflows, $W$ increases by 11%. However, a significant jump in the growth of the characteristic is observed at ng = 6. For this number of microgrooves, $W$ increases immediately by 39%. With a further increase in the number of microgrooves, $W$ increases in a weaker manner. Thus, at $n_g$ = 8, the growth is 48%, and at $n_g \geq 12$, circumferential air crossflows practically cease, providing an overall efficiency of increasing the bearing capacity from the use of longitudinal microgrooves by 61%. With other parameters, the efficiency may differ from the given estimates and is in the range of 60–75%.

The data provided testifies to the high efficiency of the use of longitudinal microgrooves to increase the bearing capacity of compressed air in the row-to-row sector of the bearing surface. If we assume that the number of microgrooves $n_g$ = 12 is optimal, then the refined estimate of their volume in relation to the volume of the bearing gap will be only 3%. Such an insignificant increase in the volume of compressed gas adjacent to the bearing gap cannot contribute to the deterioration of the dynamics of the structure; therefore, the use of microgrooves to significantly improve the static characteristics of the bearing is quite justified.

### 4.3. Modeling Circumferential Flows in End Regions

Circumferential flows can affect the reduction in load-bearing capacity and increase compliance in the end regions. Obviously, this influence will be smaller since the width of the end regions is usually smaller and, at the exit from the bearing gap, the pressure is constant and equal to the ambient pressure. At the same time, it is of interest to study the effect of the width of the end regions and eccentricity in order to assess at what width the circumferential flows have little effect on the bearing capacity.

We divide the segment $[0, L_1]$ into $n$ parts and the arc $[0, \pi]$ into $m$ parts. The basic finite-difference equations (56) will be slightly modified, where $x = j - 1$, $y = j + 1$. Due to the symmetry of the air pressure at $\varphi = 0$ and $\varphi = \pi$, we assume $x = 1$, if $x = -1$, and $y = m - 1$, if $y = m + 1$. Taking this into account, iterative Formula (56) takes the form

$$\left\{\Psi_i^j\right\}_{\alpha+1} = -\frac{1}{c_0}\left\{(b_0 + a_0 U_j)\Psi_i^y + (b_0 - a_0 U_j)\Psi_i^x + \Psi_{i+1}^j + \Psi_{i-1}^j\right\}_\alpha,$$

where $j = 0, 1, ..., m$; $i = 1, 2, \ldots, n - 1$, $\alpha$ is the iteration number.

The boundary conditions for $Z = L_1$ correspond to the conditions in (48). The boundary conditions in (66) will be used on the pressurization line $Z = 0$.

For comparison, let us find the bearing capacity of an ideal model in which there are no circumferential flows. In this case, the solution to the boundary value problem is the function

$$\Psi(Z, \varphi) = [\Pi(\varphi) - 1]\frac{L_1 - Z}{L_1} + 1.$$

The bearing load capacity can be determined numerically by the formula

$$W = \frac{2}{\pi} \int_0^{\pi} \int_0^{L_1} \left(\sqrt{\Psi} - 1\right) \cos \varphi \, dZ d\varphi.$$

The results of modeling circumferential flows on the end zones are shown in Figure 7. Graphs of the dependence of the bearing capacity $W$ on the width $L_1$ of the end zone are plotted for various values of the eccentricity $\varepsilon$. Solid lines show the dependences obtained by the iterative method. The dashed lines correspond to the ideal airflow model, where there are no circumferential flows.

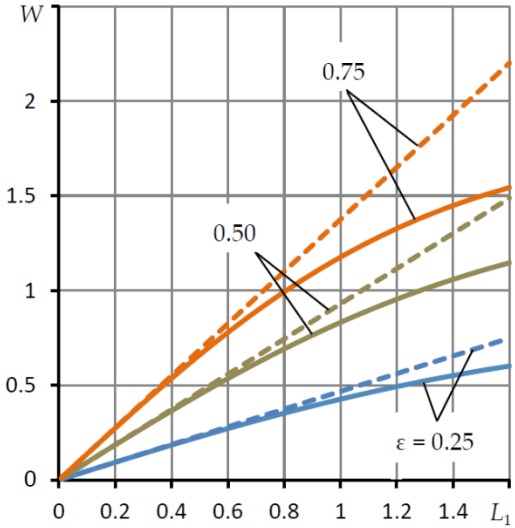

**Figure 7.** Dependencies of the load capacity $W$ of the end zones on their length $L_1$ for different eccentricities $\varepsilon$.

It can be seen from the graphs that circumferential flows affect the decrease in the bearing capacity only at $L_1 > 0.4$. Such a significant width is not typical for the location of the pressurization lines; therefore, at $L_1 \leq 0.4$, we can assume that there are no circumferential flows in the end regions.

### 4.4. Ideal Air Flow Pattern without Circumferential Crossflows

Studies have shown that if the number of microgrooves is $n_g \geq 12$ and the width of the end regions is $L_1 \leq 0.4$, then when calculating the compliance and bearing capacity, circular air flows in the bearing gap can be neglected. Let us consider two models of shaft motion—for small and arbitrary eccentricities.

#### 4.4.1. Non-Crossflowing Model for Small Eccentricities

In the absence of crossflows, the differential equation of Problem (8) takes the form

$$\frac{d^2\Delta\Psi}{dZ^2} = 0.$$

Taking this and (13) into account, the solution to the problem is the function of small pressure deviation

$$\Delta P(Z) = \Delta P_k \begin{cases} \frac{P_{k0}(L_1 - Z)}{P_0(Z)L_1}, & Z \leq L_1, \\ 1, & Z \leq L_2. \end{cases} \tag{68}$$

Substituting (68) into (14), we obtain

$$\Delta W = A_w \Delta P_k, \tag{69}$$

where

$$A_w = L_2 + \frac{P_{k0}}{L_1} \int_0^{L_1} \frac{(L_1 - Z)}{P_0(Z)} dZ.$$

The formula for the flow rate in the bearing gap near the pressurization line has the form

$$\Delta Q_h = A_{qp} \Delta P_k - A_{qe} \Delta \varepsilon, \tag{70}$$

where $A_{qp} = \frac{2P_{k0}}{L_1}$ corresponds to (30). Formulas (39) and (40) remain unchanged.

Substituting the obtained simplified Formulas (69) and (70) into (43), we obtain the formula for bearing compliance without circumferential air flows in the bearing gap for small deviations of functions from the central equilibrium position of the shaft.

Figure 8 shows the dependences of the compliance $K_e$ on the length $L$ at various values of the injection pressure $P_s$ for $\lambda_1 = 0.2$, $\chi = 0.45$, and $\varsigma = 0.15$. The solid lines show the characteristics of a conventional bearing, and the dashed lines show the bearing with longitudinal microgrooves ($n_g \geq 12$). It can be seen that at $L > 0.5$, the use of microgrooves contributes to a significant decrease in bearing compliance. Thus, at $P_s = 3$ and $L = 1.5$, the compliance decreases by 65%, and at $L = 2$, the compliance is halved. For $L > 2$, the difference will be even greater. A similar difference is observed at $P_s = 5$.

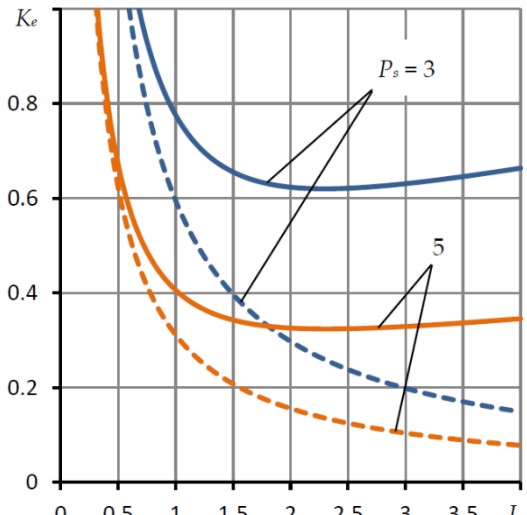

**Figure 8.** Dependences of the compliance $K_e$ of a lightly loaded bearing on the length $L$ at different values of injection pressure $P_s$, $\lambda_1 = 0.2$.

### 4.4.2. Non-Crossflowing Model for Arbitrary Eccentricities

In the absence of crossflows, the pressure and air flow rate in the lubricating gap, by analogy with (16) and (23), are determined by the formulas

$$P(Z) = \begin{cases} \sqrt{(P_k^2 - 1)\frac{L_1 - Z}{L_1} + 1}, & 0 \leq Z \leq L_1, \\ P_k, & 0 \leq Z \leq L_2, \end{cases}$$

Formulas (33) and (34) remain unchanged.

As before, we will divide the arc $\varphi \in [0, \pi]$ into $m$ parts. Let us set $\varepsilon$, and for $\varphi_j = \frac{\pi j}{m}$ $(j = 0, 1, \ldots, m)$, calculate $H$. Next, for each $j$, we solve the system of the final two nonlinear equations in (35), representing the balance of flow rates in the SECT as

$$\begin{cases} \frac{H_j^2}{L_1}\left[\left(P_k^j\right)^2 - 1\right] = A_k \, \mathrm{Brn}(P_p^j, P_k^j), \\ A_p \, \mathrm{Brn}(P_s, P_p^j) = A_k H_j \, \mathrm{Brn}(P_p^j, P_k^j) \end{cases}$$

relative to unknown pressures.

Having evaluated the pressure vector at the outlet of the annular diaphragms $P_k = \left(P_k^0, P_k^1, \ldots, P_k^m\right)$, we found the integral of the bearing capacity by the Simpson quadrature formula

$$W = \frac{2}{\pi} \int\limits_0^{2\pi} \int\limits_0^{L} (P - 1)\cos\varphi \, dZ d\varphi = \frac{2}{3m} \sum_{j=0}^{m} s_j \cos\varphi_j \left[ \frac{2L_1\left(P_k^3 - 1\right)}{3\left(P_k^2 - 1\right)} + L_2(P_k - 1) \right]_j, \quad (71)$$

where $s_j$ are the coefficients of the Simpson formula [29].

Figure 9 shows the graphs of the dependences of the bearing capacity $W$ on the eccentricity $\varepsilon$ calculated by Formula (71) for various values of the length $L$ for $P_s = 5$, $\lambda_1 = 0.2$, $\chi = 0.45$, $\varsigma = 0.15$, and $n_g \geq 12$. The graphs are similar to Figure 2.

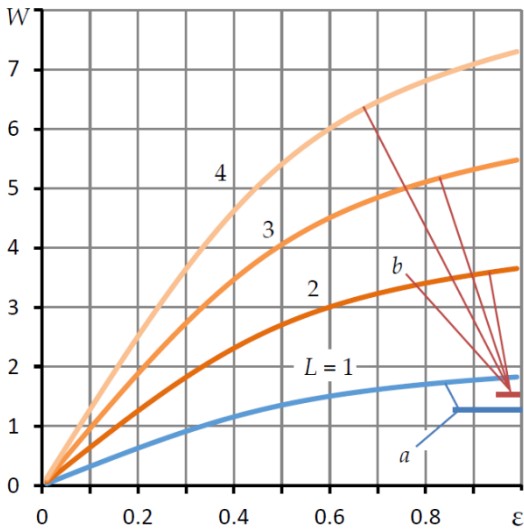

**Figure 9.** Dependences of the bearing capacity $W$ of the bearing with microgrooves on the eccentricity $\varepsilon$ at different values of length $L$.

Position $a$ in the figure indicates the maximum value of the bearing capacity $W$ of a conventional bearing of length $L = 1$, and position $b$ indicates the maximum value of $W$ curves for $L > 1$ (these curves are shown in Figure 2). With $L = 1$, the maximum bearing capacity of a bearing with microgrooves is 1.5 times greater than that of a conventional bearing; with $L = 1.5$, it is 1.8 times greater; with $L = 2$, it is 2.3 times greater; and with $L = 4$, it is 4.5 times greater. The same estimates take place for small and moderate values of eccentricity.

The data obtained show that the use of longitudinal microgrooves in the inter-row zone at $L \geq 1$ makes it possible to reduce the compliance by a factor of 1.5 and increase the bearing capacity. Moreover, the larger the relative length of the bearing, the better the performance. Therefore, the use of longitudinal microgrooves is an effective way to improve the force characteristics of a radial aerostatic bearing significantly.

## 5. Comparative Analysis of Theoretical and Experimental Data

For experimental verification of the formulas obtained, the bearing capacity of a single throttling bearing without microgrooves was found, and its compliance, stiffness, and load capacity were compared with the data of [31]. In addition, for a correct comparison, the parameter of the relative spreading of the pressurization lines $\lambda_1 = 0.5$ was adopted. Comparative data are shown in Table 1.

**Table 1.** Comparative bearing characteristics at relative eccentricity $\varepsilon = 0.5$.

| $L(l/r_0)$, $h_0$ (mm) | Supply Pressure $p_s$ (MPa) | Load Capacity (N) | | Stiffness (N·m$^{-1}$) | |
|---|---|---|---|---|---|
| | | Experiment | Theory | Experiment | Theory |
| 1.5, 0.0225 | 0.3 | 64 | 69 | $1.0 \times 10^7$ | $1.2 \times 10^7$ |
| | 0.5 | 105 | 112 | $1.5 \times 10^7$ | $1.6 \times 10^7$ |
| 1.0, 0.015 | 0.3 | 28 | 31 | $1.5 \times 10^7$ | $1.4 \times 10^7$ |
| | 0.5 | 72 | 76 | $2.0 \times 10^7$ | $1.8 \times 10^7$ |

As can be seen from the table, on the whole, the experimental and theoretical data are in satisfactory agreement. The theoretical bearing capacity is slightly higher than the experimental data. The reason is probably that the theoretical model is based on the assumption that discrete feeders can be replaced with a continuous pressurization line. As can be seen, this assumption gives an error of no more than 10%, which can be considered a completely satisfactory agreement between theory and experiment. This tendency is also observed when comparing the stiffness characteristics for $L = 1.5$. For $L = 1.0$, on the contrary, the theoretical stiffness was less than the experimental one.

Figure 10 shows the experimental values of $W$ (points) obtained by Cunningham [32] for an aerostatic bearing with two sets of orifices of the "simple diaphragm" type for $h_0 = 33 \times 10^{-6}$ mm, $n = 6$, $d = 0.56$ mm, $L = 1.5$, and $L_1 = 0$.

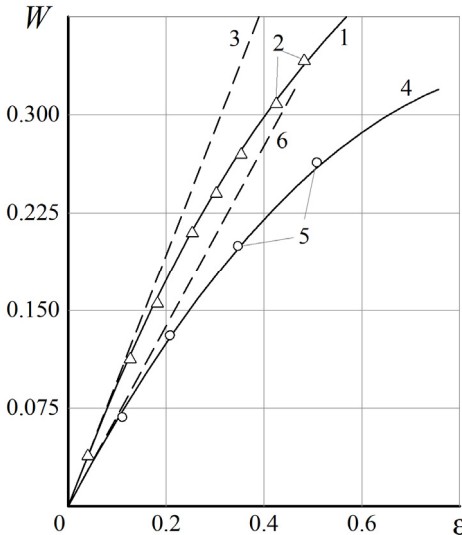

**Figure 10.** Comparison of the load characteristics of a double-row bearing.

Dependencies 1 and 4 were obtained by Shatokhin [33], 2 and 5 are points of the Cunningham experiment [32], and 3 and 6 are calculated by the formula

$$W = \frac{2(P_s - 1)\varepsilon}{\pi K}.$$

The former corresponds to $P_s = 6$, the latter $P_s = 2.3$.

Linear dependences 3 and 6, presented in Figure 10, have a satisfactory coincidence with the compared data in the region of small eccentricities ε.

## 6. Conclusions

The paper considers a double-row symmetric aerostatic radial bearing with an external combined throttling system and longitudinal microgrooves in the inter-row zone. It is hypothesized that the use of microgrooves will reduce the circumferential flows of compressed air, as a result of which the compliance should decrease and the bearing capacity should increase.

To test the hypothesis, mathematical modeling, calculations, and a theoretical study of the stationary operating modes of the bearing were carried out on the assumption that when the journal moves, the axes of the shaft and sleeve remain parallel. A study of compliance for small eccentricities of the shaft in the vicinity of the central equilibrium position of the shaft and the bearing capacity for arbitrary eccentricities was carried out. Formulas were obtained for the numerical evaluation of compliance for bearings with a smooth bushing surface and with longitudinal microgrooves. Iterative finite-difference methods for determining the fields of the squared pressure were proposed, on the basis of which the load capacity of the bearings was calculated. Experimental verification of the theoretical characteristics of the bearing was carried out, which showed a satisfactory agreement of the compared data.

The study of the compliance and load capacity of a microgroove bearing yielded impressive results. It is shown that the positive effect from the application of the improvement begins to manifest itself already at four microgrooves; at six microgrooves, it becomes significant, and at twelve or more microgrooves, circumferential flows in the bearing gap practically disappear; therefore, the load characteristics can be calculated on the basis of one-dimensional models of the longitudinal flow of air lubrication. The calculations have shown that with a length of $L = 1$, the maximum bearing capacity of a bearing with microgrooves is 1.5 times greater than that of a conventional bearing; for $L \geq 1.5$, the bearing capacity doubles or more. The result obtained allows us to recommend the proposed improvement for practical use in order to increase the load capacity of radial aerostatic bearings significantly.

It is also shown that the increase in the volume of compressed gas in the bearing gap due to the presence of microgrooves is only a few percent of a similar volume of a conventional bearing. On this basis, it was concluded that this circumstance cannot affect the deterioration of dynamic characteristics. It should be noted that bearings with combined external throttling systems have a large margin of stability, which guarantees the operability of bearings with microgrooves.

Thus, the proposed double-row aerostatic radial bearing with longitudinal microgrooves in the inter-row zone can find practical application as a better replacement for conventional aerostatic bearings in order to reduce compliance and significantly increase the bearing load capacity.

**Author Contributions:** Conceptualization, V.K.; formal analysis, A.K.; investigation, S.S., O.G., I.P. and Y.P.; data curation, M.B. and L.S.; writing—original draft preparation, V.K.; writing—review and editing, A.K.; project administration, A.K. and O.G. All authors have read and agreed to the published version of the manuscript.

**Funding:** This research received no external funding.

**Institutional Review Board Statement:** Not applicable.

**Informed Consent Statement:** Not applicable.

**Data Availability Statement:** The data presented in this study are available on request from the corresponding author. The data are not publicly available due to privacy.

**Conflicts of Interest:** The authors declare no conflict of interest.

### Nomenclature

| | |
|---|---|
| $H, h, h_0$ | dimensionless thickness, thickness of the gap, and its thickness for $\varepsilon = 0$ |
| $K_e$ | dimensionless compliance of bearing |
| $l, L$ | half of the length and dimensionless length of bearing |
| $l_1, L_1$ | half of the length and dimensionless length of end zones |
| $l_2, L_2$ | length and dimensionless length of the inter-row zones |
| $P(Z, \varphi)$ | dimensionless dynamic pressure in the bearing gap |
| $P_0(Z)$ | dimensionless static pressure in the bearing gap for $\varepsilon = 0$ |
| $p_a$ | ambient pressure |
| $p_k$ | air pressure on annular diaphragms |
| $p_p$ | air pressure in inter-throttle chambers |
| $p_s$ | supply pressure |
| $Q_h$ | dimensionless flow rate through the gap |
| $Q_k$ | dimensionless flow rate through annular damping diaphragms |
| $Q_p$ | dimensionless flow rate through simple diaphragms |
| $r_0$ | the shaft radius |
| $W$ | dimensionless bearing capacity |
| $Z$ | dimensionless longitudinal coordinate |
| $\varepsilon$ | dimensionless eccentricity |
| $\varphi$ | dimensionless circumferential coordinate |
| $\mu$ | coefficient of dynamic viscosity of the air |
| $\chi$ | normalized adjustment coefficient of the external throttling system |
| $\varsigma$ | normalized adjustment coefficient of annular damping diaphragms |

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
