# Peer review of "Numerical Modeling on the Compliance and Load Capacity of a Two-Row Aerostatic Journal Bearing with Longitudinal Microgrooves in the Inter-Row Zone"

_applsci, doi:10.3390/app11125714_

Round 1
Reviewer 1 Report
Manuscript ID: applsci-1245151
Title: Numerical modeling on the compliance and load capacity of a two-row aerostatic journal bearing with longitudinal microgrooves in the inter-row zone
Authors: Vladimir Aleksandrovich Kodnyanko, Stanislav Shatokhin, Andrey Kurzakov, Yuri Pikalov, Iakov Pikalov, Olga Grigorieva, Lilia Strok, Maxim Brungardt
Dear authors,
Air bearings are one of the attracting machine elements for particular applications. This is an intensive simulation work on the surface texturing of mating surfaces. The work is publishable and I recommend improving the quality for readers’ benefit.
- The relation between simulation results and the real world is of importance. Table 1 shows the results without surface texturing from conference abstract, which is not always easy to access. Therefore, brief description regarding experimental setup and conditions are recommended. Also the readers are interested in experiments with surface texture. Citing relevant literature will be fine.
- X and Y axis in Figures 2,3,6-9 are necessary. It needs the unit for real measure or clarify whether relative value or not.
- The model considers the viscosity of the operating gas. In the real experiment, humidity influences the lubrication performances. It needs to discuss whether other physical properties such as compress properties, heat capacity, and other properties should be considered. Note that the operating gas does not always obey the rule of ideal gas under these conditions.
Author Response
Author's Reply to the Review Report (Reviewer 1)
- The relation between simulation results and the real world is of importance. Table 1 shows the results without surface texturing from conference abstract, which is not always easy to access. Therefore, brief description regarding experimental setup and conditions are recommended. Also the readers are interested in experiments with surface texture. Citing relevant literature will be fine.
Answer 1.
We provide a direct link to the article where the experimental setup is described in detail. The results of the experiments carried out and their description are also given there:
https://www.researchgate.net/publication/228949539_Dynamic_properties_of_aerostatic_journal_bearings
- X and Y axis in Figures 2,3,6-9 are necessary. It needs the unit for real measure or clarify whether relative value or not.
Answer 2.
The axes of the graphs have labels for the values. However, we did not specify any units because both characteristics are dimensionless.
- The model considers the viscosity of the operating gas. In the real experiment, humidity influences the lubrication performances. It needs to discuss whether other physical properties such as compress properties, heat capacity, and other properties should be considered. Note that the operating gas does not always obey the rule of ideal gas under these conditions.
Answer 3.
In the presented theoretical studies, the viscosity of the gas and it’s temperature were considered constant, and the air flow in the thin bearing layer was assumed to be laminar. It was also assumed that air is considered a compressed medium not only in large volumes, but also in thin lubricating layers of a bearing. In the theory of gas lubrication, such assumptions are generally accepted.
Reviewer 2 Report
- The current study evaluates the load carrying capacity of a aerostatic journal bearing. For this the authors conduct an FE model which simulate a double-row aerostatic journal bearing that have a throttling system and longitudinal microgrooves in the inter-row zone. The authors predict that these microgrooves have some potential in reducing the compressed air flow to improve the compliance. The authors use the FE model to make some recommendation for improved load capacity bearing for aerostatic applications.
- Please avoid using we, us, our..etc in the abstract and everywhere else in the manuscript.
- Please consider reviewing the abstract and highlight the novelty, major findings and conclusions.
- There is abuse of citations in the introduction [1-8] [13-16] [19-24] 16 references in 8 lines? This is not acceptable and the authors should give full credit to those referenced articles somewhere in the manuscript.
- Section 3 can the authors only keep the main equations in this section and move the remaining parts into an appendix, there is quiet few formulas and detailed discussion about them which is more suitable for a thesis but not a journal publication
- What are the limitations in your FE model, also what are the limitations/improvements compared to previous models reported in the open literature
- I am unable to find where does the introduction ends and where does the results and discussion starts? Please use the regular journal template format instead of the current one.
- Somewhere near the end of your introduction please answer the following question: What is the research gap did you find from the previous researchers in your field? Mention it properly. It will improve the strength of the article.
- “can be approximately considered constant and equal to the air pressure at the outlet of the nearest annular diaphragm.” Please support this claim with a reference
- “It can be seen from the graphs that circumferential flows affect the decrease in the bearing capacity only at L1 > 0.4. Such a significant width is not typical for the location of the pressurization lines, therefore, at L1 ≤ 0.4 we can assume that there are no circumferential flows in the end regions.” How about previous studies did they report similar observation like yours or different. Please try to compare your most important findings with any available literature similar to your work
- I am a bit confused, in your model you include the T as the absolute air temperature but did your model account for temperature changes? i.e. does it calculate how air pressure changes due to change of temperature or it does not account for it?
- In section 5 “5. Comparative analysis of theoretical and experimental data “ The results are merely described and is limited to comparing the experimental observation. The authors are encouraged to include more detailed discussion and critically discuss the observations from this investigation with existing literature.
Author Response
Author's Reply to the Review Report (Reviewer 2)
- The current study evaluates the load carrying capacity of a aerostatic journal bearing. For this the authors conduct an FE model which simulate a double-row aerostatic journal bearing that have a throttling system and longitudinal microgrooves in the inter-row zone. The authors predict that these microgrooves have some potential in reducing the compressed air flow to improve the compliance. The authors use the FE model to make some recommendation for improved load capacity bearing for aerostatic applications.
-
- Please avoid using we, us, our..etc in the abstract and everywhere else in the manuscript.
Answer 2.
The authors consider this style of presentation of the article to be quite acceptable. There are many examples of articles that use the same style. We give an example of such an article.
https://www.academia.edu/1472203/2003_A_Superfast_Method_for_Solving_Toeplitz_Linear_Least_Squares_Problems
- Please consider reviewing the abstract and highlight the novelty, major findings and conclusions.
Answer 3.
In our opinion, the abstract contains all the main provisions mentioned by the reviewer.
- There is abuse of citations in the introduction [1-8] [13-16] [19-24] 16 references in 8 lines? This is not acceptable and the authors should give full credit to those referenced articles somewhere in the manuscript.
Answer 4.
The authors do not consider such links to be abusive. The reviewer expresses his preferences, but there are many works where there are examples of such links.
A link to a similar article is given above.
- Section 3 can the authors only keep the main equations in this section and move the remaining parts into an appendix, there is quiet few formulas and detailed discussion about them which is more suitable for a thesis but not a journal publication
Answer 5.
If we act in accordance with the advices of the reviewer, then this would violate the logic of the presentation of the article material, in addition, this would require significant revision of it, which is unacceptable under these conditions, besides, we are sure that there is no special need for this.
- What are the limitations in your FE model, also what are the limitations/improvements compared to previous models reported in the open literature
Answer 6.
In this article, the generally accepted restrictions are used, according to which the air flow in thin gaps is laminar, the temperature and viscosity of the air are constant, a number of other restrictions are used, as mentioned in the article. Improvements relate to the studied bearing design, compliance and load capacity, applied mathematical and computational techniques, which are described in the article.
- I am unable to find where does the introduction ends and where does the results and discussion starts? Please use the regular journal template format instead of the current one.
Answer 7.
We cannot use the standard format of the article, because this would require significant revision of it, which is not feasible within the allotted time. The introduction section ends where Chapter 2 “Characteristics of a conventional aerostatic bearing” begins.
- Somewhere near the end of your introduction please answer the following question: What is the research gap did you find from the previous researchers in your field? Mention it properly. It will improve the strength of the article.
Answer 8.
We did not find any research gaps from previous researchers, as we are investigating new types of structures that have no analogues in the world. Therefore, the constructions and methods used are simply nothing to compare with.
- “can be approximately considered constant and equal to the air pressure at the outlet of the nearest annular diaphragm.” Please support this claim with a reference
Answer 9.
We agree with the comment of the reviewer. We have added the corresponding link to the list of references.
- “It can be seen from the graphs that circumferential flows affect the decrease in the bearing capacity only at L1 > 0.4. Such a significant width is not typical for the location of the pressurization lines, therefore, at L1 ≤ 0.4 we can assume that there are no circumferential flows in the end regions.” How about previous studies did they report similar observation like yours or different. Please try to compare your most important findings with any available literature similar to your work
Answer 10.
Similar observations were carried out earlier by other authors. There is a corresponding link in the article.
- I am a bit confused, in your model you include the T as the absolute air temperature but did your model account for temperature changes? i.e. does it calculate how air pressure changes due to change of temperature or it does not account for it?
Answer 11.
In our research, we assumed that the temperature of the lubricant in the bearing flow paths is constant. This is the generally accepted basis.
- In section 5 “5. Comparative analysis of theoretical and experimental data “The results are merely described and is limited to comparing the experimental observation. The authors are encouraged to include more detailed discussion and critically discuss the observations from this investigation with existing literature.
Answer 12.
In this case, we compare theoretical and experimental data for a rigidly suspended bearing in order to substantiate the coincidence of characteristics for particular cases. Our work is theoretical. A complete experimental study of the structure is planned to be carried out after the publication of this material, which largely gives recommendations for the optimal parameters of the experimental setup.
Round 2
Reviewer 1 Report
Dear authors,
Thank you for revision. In my opinion, this is acceptable.
Author Response
Dear editor.
I do not see the comments of the reviewer.
Best regards, Vladimmir Kodnyanko
Reviewer 2 Report
All questions were answered. the paper can be accepted for publication
Congratulations to the authors
Author Response

(The authors gave the same response as above.)
